# Training deep quantum neural networks

Kerstin Beer [1]*, Dmytro Bondarenko[1], Terry Farrelly [1,2], Tobias J. Osborne[1], Robert Salzmann[1,3], Daniel Scheiermann[1] & Ramona Wolf [1]

Neural networks enjoy widespread success in both research and industry and, with the advent of quantum technology, it is a crucial challenge to design quantum neural networks for fully quantum learning tasks. Here we propose a truly quantum analogue of classical neurons, which form quantum feedforward neural networks capable of universal quantum computation. We describe the efficient training of these networks using the fidelity as a cost function, providing both classical and efficient quantum implementations. Our method allows for fast optimisation with reduced memory requirements: the number of qudits required scales with only the width, allowing deep-network optimisation. We benchmark our proposal for the quantum task of learning an unknown unitary and find remarkable generalisation behaviour and a striking robustness to noisy training data.

[1] Institut für Theoretische Physik, Leibniz Universität Hannover, Appelstraße 2, 30167 Hannover, Germany. [2] ARC Centre for Engineered Quantum Systems, School of Mathematics and Physics, University of Queensland, Brisbane, QLD 4072, Australia. [3] Department of Applied Mathematics and Theoretical Physics, University of Cambridge, Cambridge CB3 0WA, UK. *email: kerstin.beer@itp.uni-hannover.de

Machine learning (ML), particularly applied to deep neural networks via the backpropagation algorithm, has enabled a wide spectrum of revolutionary applications ranging from the social to the scientific[1,2]. Triumphs include the now everyday deployment of handwriting and speech recognition through to applications at the frontier of scientific research[2–4]. Despite rapid theoretical and practical progress, ML training algorithms are computationally expensive and, now that Moore's law is faltering, we must contemplate a future with a slower rate of advance[5]. However, new exciting possibilities are opening up due to the imminent advent of quantum computing devices that directly exploit the laws of quantum mechanics to evade the technological and thermodynamical limits of classical computation[5].

The exploitation of quantum computing devices to carry out quantum maching learning (QML) is in its initial exploratory stages[6]. One can exploit classical ML to improve quantum tasks ("QC" ML, see refs. [7,8] for a discussion of this terminology) such as the simulation of many-body systems[9], adaptive quantum computation[10] or quantum metrology[11], or one can exploit quantum algorithms to speed up classical ML ("CQ" ML)[12–15], or, finally, one can exploit quantum computing devices to carry out learning tasks with quantum data ("QQ" ML)[16–24]. A good review on this topic can be found in ref. [25]. Particularly relevant to the present work is the recent paper of Verdon et al.[26], where quantum learning of parametrised unitary operations is carried out coherently. The task of learning an unknown unitary was also studied in a different setting in ref. [27], where the authors focussed on storing the unitary in a quantum memory while having a limited amount of resources. This was later generalised to probabilistic protocols in ref. [28]. There are still many challenging open problems left for QML, particularly, the task of developing quantum algorithms for learning tasks involving quantum data.

A series of hurdles face the designer of a QML algorithm for quantum data. These include, finding the correct quantum generalisation of the perceptron, (deep) neural network architecture, optimisation algorithm, and loss function. In this paper we meet these challenges and propose a natural quantum perceptron which, when integrated into a quantum neural network (QNN), is capable of carrying out universal quantum computation. Our QNN architecture allows for a quantum analogue of the classical backpropagation algorithm by exploiting completely positive layer transition maps. We apply our QNN to the task of learning an unknown unitary, both with and without errors. Our classical simulation results are very promising and suggest the feasibility of our procedure for noisy intermediate scale (NISQ) quantum devices, although one would still have to study how noise in the network itself influences the performance.

There are now several available quantum generalisations of the perceptron, the fundamental building block of a neural network[1,2,29–35]. In the context of CQ learning (in contrast to QQ learning, which we consider here) proposals include refs. [36–40], where the authors exploit a qubit circuit setup, though the gate choices and geometry are somewhat more specific than ours. Another interesting approach is to use continuous-variable quantum systems (e.g., light) to define quantum perceptrons[41–43].

With the aim of building a fully quantum deep neural network capable of universal quantum computation we have found it necessary to modify the extant proposals somewhat. In this paper we define a quantum perceptron to be a general unitary operator acting on the corresponding input and output qubits, whose parameters incorporate the weights and biases of previous proposals in a natural way. Furthermore, we propose a training algorithm for this quantum neural network that is efficient in the sense that it only depends on the width of the individual layers and not on the depth of the network. It is also an important

observation that there is no barren plateau in the cost function landscape. We find that the proposed network has some remarkable properties, as the ability to generalise from very small data sets and a remarkable tolerance to noisy training data.

## Results

**The network architecture.** The smallest building block of a quantum neural network is the quantum perceptron, the quantum analogue of perceptrons used in classical machine learning. In our proposal, a quantum perceptron is an arbitrary unitary operator with $m$ input qubits and $n$ output qubits. Our perceptron is then simply an arbitrary unitary applied to the $m + n$ input and output qubits which depends on $(2^{m+n})^2 - 1$ parameters. The input qubits are initialised in a possibly unknown mixed state $\rho^{\text{in}}$ and the output qubits in a fiducial product state $|0 \cdots 0\rangle_{\text{out}}$ (note that this scheme can easily be extended to qudits). For simplicity in the following we focus on the case where our perceptrons act on $m$ input qubits and one output qubit, i.e., they are $(m + 1)$-qubit unitaries.

Now we have a quantum neuron which can describe our quantum neural network architecture. Motivated by analogy with the classical case and consequent operational considerations we propose that a QNN is a quantum circuit of quantum perceptrons organised into $L$ hidden layers of qubits, acting on an initial state $\rho^{\text{in}}$ of the input qubits, and producing an, in general, mixed state $\rho^{\text{out}}$ for the output qubits according to

$$\rho^{\text{out}} \equiv \text{tr}_{\text{in,hid}}\left(\mathcal{U}(\rho^{\text{in}} \otimes |0 \cdots 0\rangle_{\text{hid,out}}\langle 0 \cdots 0|)\mathcal{U}^{\dagger}\right), \quad (1)$$

where $\mathcal{U} \equiv U^{\text{out}}U^{L}U^{L-1} \cdots U^{1}$ is the QNN quantum circuit, $U^{l}$ are the layer unitaries, comprised of a product of quantum perceptrons acting on the qubits in layers $l-1$ and $l$. It is important to note that, because our perceptrons are arbitrary unitary operators, they do not, in general, commute, so that the order of operations is significant. See Fig. 1 for an illustration.

It is a direct consequence of the quantum-circuit structure of our QNNs that they can carry out universal quantum computation, even for two-input one-output qubit perceptrons. More remarkable, however, is the observation that a QNN comprised of quantum perceptrons acting on 4-level qudits that commute within each layer, is still capable of carrying out universal quantum computation (see Supplementary Note 1 and Supplementary Fig. 1 for details). Although commuting qudit perceptrons suffice, we have actually found it convenient in

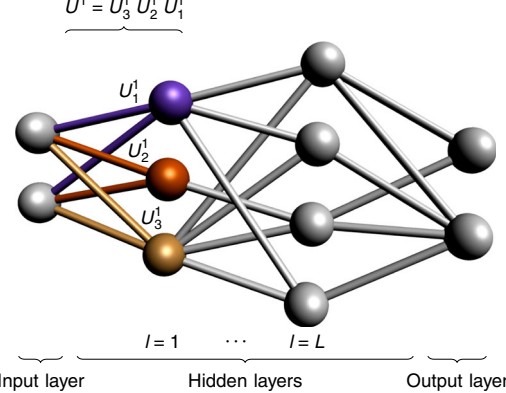

**Fig. 1 A general quantum feedforward neural network.** A quantum neural network has an input, output, and $L$ hidden layers. We apply the perceptron unitaries layerwise from top to bottom (indicated with colours for the first layer): first the violet unitary is applied, followed by the orange one, and finally the yellow one.

practice to exploit noncommuting perceptrons acting on qubits. In fact, the most general form of our quantum perceptrons can implement any quantum channel on the input qudits (see Supplementary Fig. 2), so one could not hope for any more general notion of a quantum perceptron.

A crucial property of our QNN definition is that the network output may be expressed as the composition of a sequence of completely positive layer-to-layer transition maps $\mathcal{E}^l$:

$$\rho^{\text{out}} = \mathcal{E}^{\text{out}}\big(\mathcal{E}^L\big(\ldots \mathcal{E}^2\big(\mathcal{E}^1\big(\rho^{\text{in}}\big)\big)\ldots\big)\big), \qquad (2)$$

where $\mathcal{E}^l\big(X^{l-1}\big) \equiv \text{tr}_{l-1}\big(\prod_{j=m_l}^1 U_j^l\big(X^{l-1}\otimes|0\cdots0\rangle_l\langle0\cdots0|\big)$ $\prod_{j=1}^{m_l} U_j^{l\dagger}\big)$, $U_j^l$ is the $j$th perceptron acting on layers $l-1$ and $l$, and $m_l$ is the total number of perceptrons acting on layers $l-1$ and $l$. This characterisation of the output of a QNN highlights a key structural characteristic: information propagates from input to output and hence naturally implements a quantum feedforward neural network. This key result is the fundamental basis for our quantum analogue of the backpropagation algorithm.

As an aside, we can justify our choice of quantum perceptron for our QNNs, by contrasting it with a recent notion of a quantum perceptron as a controlled unitary[36,44], i.e., $U = \sum_\alpha|\alpha\rangle\langle\alpha| \otimes U(\alpha)$, where $|\alpha\rangle$ is some basis for the input space and $U(\alpha)$ are parametrised unitaries. Substituting this definition into Eq. (2) implies that the output state is the result of a measure-and-prepare, or cq, channel. That is, $\rho^{\text{out}} = \sum_\alpha\langle\alpha|\rho^{\text{in}}|\alpha\rangle U(\alpha)|0\rangle\langle0|U(\alpha)^\dagger$. Such channels have no nonzero quantum channel capacity and cannot carry out general quantum computation.

**The training algorithm.** Now that we have an architecture for our QNN we can specify the learning task. Here, it is important to be clear about what part of the classical scenario we quantize. One possibility is to replace each classical sample of an unknown underlying probability distribution by a different quantum state. Hence, in the quantum setting, the underlying probability distribution will then be a distribution over quantum states. The second possibility is to identify the distribution itself with a quantum state, which we assume in this work, in which case it is justified to say that $N$ samples correspond to $N$ identical quantum states. We focus on the scenario where we have repeatable access to training data in the form of pairs $\big(|\phi_x^{\text{in}}\rangle, |\phi_x^{\text{out}}\rangle\big)$, $x = 1, 2, \ldots, N$, of possibly unknown quantum states. (It is crucial that we can request multiple copies of a training pair $\big(|\phi_x^{\text{in}}\rangle, |\phi_x^{\text{out}}\rangle\big)$ for a specified $x$ in order to overcome quantum projection noise in evaluating the derivative of the cost function.) Furthermore, the number of copies per training round needed grows quickly with the number of neurons (linearly with the number of network parameters), i.e., $n_{\text{proj}} \times n_{\text{params}}$, where $n_{\text{proj}}$ is the factor coming from repetition of measurements to reduce projection noise, and $n_{\text{params}}$ is the total number of parameters in the network given by $\sum_{l=1}^{L+1}(4^{(m_{l-1}+1)} - 1) \times m_l$, where $m_l$ is the number of perceptrons acting on layers $l-1$ and layer $l$, and the $-1$ term appears because the overall phase of the unitaries is unimportant. See Supplementary Note 5 for more details and a comparison to state tomography. This means that in the near term, for large networks, only sparsely connected networks may be practical for experimental purposes. An exception would be if the problem being considered is such that the training data is easy to produce, e.g., if the output states are produced by allowing input states to thermalize by simply interacting with environment, thus producing the output states. For concreteness from now on we focus on the restricted case where $|\phi_x^{\text{out}}\rangle = V|\phi_x^{\text{in}}\rangle$, where $V$ is some unknown unitary operation. This scenario is typical when one has

access to an untrusted or uncharacterised device which performs an unknown quantum information processing task and one is able to repeatably initialise and apply the device to arbitrary initial states.

To evaluate the performance of our QNN in learning the training data, i.e., how close is the network output $\rho_x^{\text{out}}$ for the input $|\phi_x^{\text{in}}\rangle$ to the correct output $|\phi_x^{\text{out}}\rangle$, we need a cost function. Operationally, there is an essentially unique measure of closeness for (pure) quantum states, namely the fidelity, and it is for this reason that we define our cost function to be the fidelity between the QNN output and the desired output averaged over the training data:

$$C = \frac{1}{N}\sum_{x=1}^N \langle\phi_x^{\text{out}}|\rho_x^{\text{out}}|\phi_x^{\text{out}}\rangle. \qquad (3)$$

Note that the cost function is a direct generalisation of the risk function considered in training calssical deep networks and we can efficiently simulate it. Also note that it takes a slightly more complicated form when the training data output states are not pure (in that case, we simply use the fidelity for mixed states: $F(\rho, \sigma) := \big[\text{tr}\sqrt{\rho^{1/2}\sigma\rho^{1/2}}\big]^2$), which may occur if we were to train our network to learn a quantum channel.

The cost function varies between 0 (worst) and 1 (best). We train the QNN by optimising the cost function $C$. This, as in the classical case, proceeds via update of the QNN parameters: at each training step, we update the perceptron unitaries according to $U \to e^{i\epsilon K}U$, where $K$ is the matrix that includes all parameters of the corresponding perceptron unitary and $\epsilon$ is the chosen step size. The matrices $K$ are chosen so that the cost function increases most rapidly: the change in $C$ is given by

$$\Delta C = \frac{\epsilon}{N}\sum_{x=1}^N\sum_{l=1}^{L+1} \text{tr}\big(\sigma_x^l \Delta\mathcal{E}^l\big(\rho_x^{l-1}\big)\big), \qquad (4)$$

where $L+1 = \text{out}$, $\rho_x^l = \mathcal{E}^l\big(\cdots\mathcal{E}^2\big(\mathcal{E}^1\big(\rho^{\text{in}}\big)\big)\cdots\big)$, $\sigma_x^l = \mathcal{F}^{l+1}\big(\cdots\mathcal{F}^L\big(\mathcal{F}^{\text{out}}\big(|\phi_x^{\text{out}}\rangle\langle\phi_x^{\text{out}}|\big)\big)\cdots\big)$, and $\mathcal{F}(X) \equiv \sum_\alpha A_\alpha^\dagger X A_\alpha$ is the adjoint channel for the CP map $\mathcal{E}(X) = \sum_\alpha A_\alpha X A_\alpha^\dagger$. From Eq. (4), we obtain a formula for the parameter matrices (this is described in detail in Supplementary Note 2). At this point, the layer structure of the network comes in handy: To evaluate $K_j^l$ for a specific perceptron, we only need the output state of the previous layer, $\rho^{l-1}$ (which is obtained by applying the layer-to-layer channels $\mathcal{E}^1, \mathcal{E}^2 \ldots \mathcal{E}^{l-1}$ to the input state), and the state of the following layer $\sigma^l$ obtained from applying the adjoint channels to the desired output state up to the current layer (see Box 1). A striking feature of this algorithm is that the parameter matrices may be calculated layer-by-layer without ever having to apply the unitary corresponding to the full quantum circuit on all the constituent qubits of the QNN in one go. In other words, we need only access two layers at any given time, which greatly reduces the memory requirements of the algorithm. Hence, the size of the matrices in our calculation only scales with the width of the network, enabling us to train deep QNNs.

**Simulation of learning tasks.** It is impossible to classically simulate deep QNN learning algorithms for more than a handful of qubits due to the exponential growth of Hilbert space. To evaluate the performance of our QML algorithm we have thus been restricted to QNNs with small widths. We have carried out pilot simulations for input and output spaces of $m = 2$ and 3 qubits and have explored the behaviour of the QML gradient descent algorithm for the task of learning a random unitary $V$

---

**Box 1 | Training algorithm**

**1. Initialize:**
Choose the initial $U_j^l$ randomly for all $j$ and $l$.

**2. Feedforward:** For every training pair $\left(\left|\phi_x^{\text{in}}\right\rangle, \left|\phi_x^{\text{out}}\right\rangle\right)$ and every layer $l$, perform the following steps:
**2a.** Apply the channel $\mathcal{E}^l$ to the output state of layer $l-1$: Tensor $\rho_x^{l-1}$ with layer $l$ in state $|0\ldots0\rangle_l$ and apply $U^l = U_{m_l}^l \ldots U_1^l$:

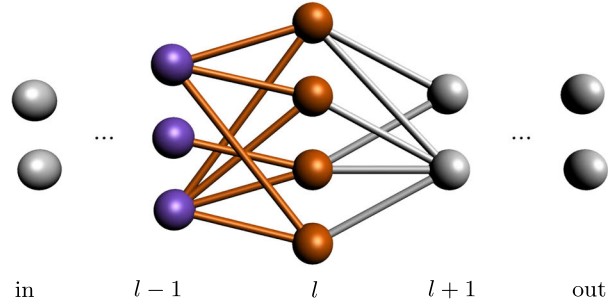

in $l-1$ $l$ $l+1$ out

**2b.** Trace out layer $l-1$ and store $\rho_x^l$.

**3. Update the network:**
**3a.** Calculate the parameter matrices given by

$$K_j^l = \eta \frac{2^{m_{l-1}}}{N} \sum_{x=1}^{N} \text{tr}_{\text{rest}} M_j^l$$

where the trace is over all qubits that are not affected by $U_j^l$, $\eta$ is the learning rate and

$$M_j^l = \left[ \prod_{\alpha=j}^{1} U_\alpha^l \left(\rho_x^{l-1,l}\right) \prod_{\alpha=1}^{j} U_\alpha^{l\,\dagger}, \prod_{\alpha=j+1}^{m_l} U_\alpha^{l\,\dagger} \left(\mathbb{I}_{l-1} \otimes \sigma_x^l\right) \prod_{\alpha=m_l}^{j+1} U_\alpha^l \right],$$

where $\rho_x^{l-1,l} = \rho_x^{l-1} \otimes |0\ldots0\rangle_l \langle0\ldots0|$,
$\sigma_x^l = \mathcal{F}^{l+1}\left(\ldots\mathcal{F}^{\text{out}}\left(\left|\phi_x^{\text{out}}\right\rangle\left\langle\phi_x^{\text{out}}\right|\right)\ldots\right)$ and $\mathcal{F}^l$ is the adjoint channel to $\mathcal{E}^l$, i.e. the transition channel from layer $l+1$ to layer $l$. Below, the two parts of the commutator are depicted:

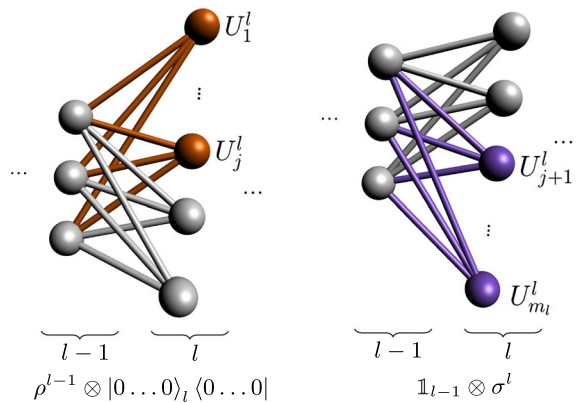

$\underbrace{\quad}_{l-1} \underbrace{\quad}_{l}$ $\underbrace{\quad}_{l-1} \underbrace{\quad}_{l}$

$\rho^{l-1} \otimes |0\ldots0\rangle_l \langle0\ldots0|$ $\mathbb{1}_{l-1} \otimes \sigma^l$

**3b.** Update each unitary $U_j^l$ according to $U_j^l \rightarrow e^{i\epsilon K_j^l} U_j^l$.

**4. Repeat:** Repeat step 2. and 3. until the cost function reaches its maximum.

---

(see Supplementary Note 4 and Supplementary Figs. 4–6 for the implementation details). We focussed on two separate tasks: In the first task we studied the ability of a QNN to generalise from a limited set of random training pairs $\left(\left|\phi_x^{\text{in}}\right\rangle, V\left|\phi_x^{\text{in}}\right\rangle\right)$, with $x=1$, ..., $N$, where $N$ was smaller than the Hilbert space dimension. The results are displayed in Fig. 2a. Here we have plotted the (numerically obtained) cost function after training alongside a theoretical estimate of the optimal cost function for the best unitary possible which exploits all the available information (for which $C \sim \frac{n}{N} + \frac{N-n}{ND(D+1)}\left(D + \min\{n^2+1, D^2\}\right)$), where $n$ is the number of training pairs, $N$ the number of test pairs and $D$ the Hilbert space dimensions). Here we see that the QNN matches the theoretical estimate and demonstrates the remarkable ability of our QNNs to generalise.

The second task we studied was aimed at understanding the robustness of the QNN to corrupted training data (e.g., due to decoherence). To evaluate this we generated a set of $N$ good training pairs and then corrupted $n$ of them by replacing them with random quantum data, where we chose the subset that was replaced by corrupted data randomly each time. We evaluated the cost function for the good pairs to check how well the network has learned the actual unitary. As illustrated in Fig. 2b the QNN is extraordinarily robust to this kind of error.

A crucial consequence of our numerical investigations was the absence of a "barren plateau" in the cost function landscape for our QNNs[45]. There are two key reasons for this: firstly, according to McClean et al.[45], "The gradient in a classical deep neural network can vanish exponentially in the number of layers [...], while in a quantum circuit the gradient may vanish exponentially in the number of qubits." This point does not apply to our QNNs because the gradient of a weight in the QNN does not depend on all the qubits but rather only on the number of paths connecting that neuron to the output, just as it does classically.

(This is best observed in the Heisenberg picture.) Thus, indeed, the gradient vanishes exponentially in the number of layers, but not in the number of qubits. Secondly, our cost function differs from that of McClean et al.[45]: they consider energy minimisation of a local hamiltonian, whereas we consider a quantum version of the risk function. Our quantity is not local, and this means that Levy's lemma-type argumentation does not directly apply. In addition, we always initialised our QNNs with random unitaries and we did not observe any exponential reduction in the value of the parameter matrices $K$ (which arise from the derivative of our QNN with respect to the parameters). This may be intuitively understood as a consequence of the nongeneric structure of our QNNs: at each layer we introduce new clean ancilla, which lead to in general, dissipative output.

## Discussion

The QNN and training algorithm we have presented here lend themselves well to the coming era of NISQ devices. The network architecture enables a reduction in the number of coherent qubits required to store the intermediate states needed to evaluate a QNN. Thus we only need to store a number of qubits scaling with the width of the network. This remarkable reduction does come at a price, namely, we require multiple evaluations of the network to estimate the derivative of the cost function. However, in the near term, this tradeoff is a happy one as many NISQ architectures—most notably superconducting qubit devices—can easily and rapidly repeat executions of a quantum circuit. For example, the recently reported experiment involving the "Sycamore" quantum computer executed one instance of a quantum circuit a million times in 200 s[46]. It is the task of adding coherent qubits that will likely be the challenging one in the near term and working with this constraint is the main goal here. A crucial problem that has to be taken into account with regard to NISQ devices is the

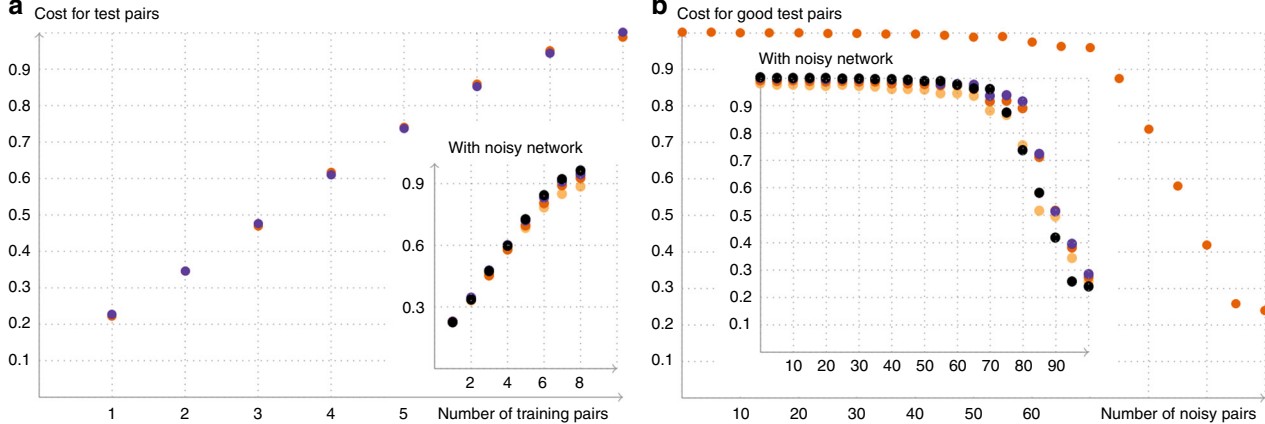

**Fig. 2 Numerical results.** In both plots, the insets show the behaviour of the quantum neural network under approximate depolarizing noise. The colours indicate the strength $t$ of the noise: black $t = 0$, violet $t = 0.0033$, orange $t = 0.0066$, yellow $t = 0.01$. For a more detailed discussion of the noise model see Supplementary Note 3 and Supplementary Fig. 3. Panel (**a**) shows the ability of the network to generalize. We trained a 3-3-3 network with $\epsilon = 0.1$, $\eta = 2/3$ for 1000 rounds with $n = 1, 2, ..., 8$ training pairs and evaluated the cost function for a set of 10 test pairs afterwards. We averaged this over 20 rounds (orange points) and compared the result with the estimated value of the optimal achievable cost function (violet points). Panel (**b**) shows the robustness of the QNN to noisy data. We trained a 2-3-2 network with $\epsilon = 0.1$, $\eta = 1$ for 300 rounds with 100 training pairs. In the plot, the number on the $x$-axis indicates how many of these pairs were replaced by a pair of noisy (i.e. random) pairs and the cost function is evaluated for all "good" test pairs.

inevitable noise within the device itself. Interestingly, we have obtained numerical evidence that, for approximate depolarising noise, QNNs are robust (see inset of Fig. 2).

In this paper we have introduced natural quantum generalisations of perceptrons and (deep) neural networks, and proposed an efficient quantum training algorithm. The resulting QML algorithm, when applied to our QNNs, demostrates remarkable capabilities, including, the ability to generalise, tolerance to noisy training data, and an absence of a barren plateau in the cost function landscape. There are many natural questions remaining in the study of QNNs including generalising the quantum perceptron definition further to cover general CP maps (thus incorporating a better model for decoherence processes), studying the effects of overfitting, and optimised implementation on the next generation of NISQ devices.

## Data availability

All results were obtained using Mathematica and Matlab. The code is available at https://github.com/qigitphannover/DeepQuantumNeuralNetworks.

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

## Acknowledgements

Helpful correspondence and discussions with Lorenzo Cardarelli, Polina Feldmann, Andrew Green, Alexander Hahn, Amit Jamadagni, Maria Kalabakov, Sebastian Kinnewig, Roger Melko, Laura Niermann, Simone Pfau, Marvin Schwiering, Deniz E. Stiegemann and E. Miles Stoudenmire are gratefully acknowledged. This work was supported by the DFG through SFB 1227 (DQ-mat), the RTG 1991, and Quantum Frontiers. T.F. was supported by the Australian Research Council Centres of Excellence for Engineered Quantum Systems (EQUS, CE170100009). The publication of this article was funded by the Open Access Fund of the Leibniz Universität Hannover.

## Author contributions

This project was conceived of, and initiated in, discussions of T.J.O. and D.B. The QNN architecture was formulated by T.J.O., T.F., R.W. and K.B. Operational considerations were investigated by D.B. Classical numerical implementations and investigations were developed by R.S. and R.W. Universality of the QNN model was discovered by T.F. The quantum implementation was developed by K.B. D.S. investigated the behaviour of the QNN under noise. All authors contributed to writing the paper.

## Competing interests

The authors declare no competing interests.
