## [Peer Review File · Nature Communications]

Reviewers' Comments:

Reviewer #1:

Remarks to the Author:

The authors propose an architecture for a feed-forward quantum neural network (QNN) that works with quantum data. Neurons are defined as qudits, and perceptrons (the mechanism to update the state of output neurons based on the state of input neurons) are modelled by arbitrary unitary operations acting on $m+1$ quantum systems (m inputs and 1 output). The propagation of information through the network can be understood as a sequence of completely positive maps from one layer of neurons through the next. This structure, which is quite generic, allows for a rather straightforward quantum formalization of the backpropagation algorithm used for training classical feed-forward neural networks. In addition, because the process of updating the state of one layer only depends on the state of the previous layer, one would not need to have simultaneous control of all neurons in the network in an experimental implementation. This is a nice feature that permits reusing quantum degrees of freedom from already processed layers as new upcoming layers reinitialized in a $|0\rangle$ state. Moreover, the authors show how their QNN is capable of universal quantum computation.

To demonstrate the functioning of their QNN, the authors consider the problem of learning a random unitary transformation V given access to quantum training data in the form of pairs $(|\psi_{in,x}\rangle, V|\psi_{in,x}\rangle)$. The QNN is trained by maximizing the fidelity between the state of the output layer $|\psi_{out,x}\rangle$ with the true output state $V|\psi_{in,x}\rangle$. This is done via backpropagation by updating the parameters of all unitaries (perceptrons) over many rounds per training state. The authors show that the QNN is able to successfully learn the unknown unitary, with notable generalization performance, and in a way that is very robust against corrupted training data (random pairs of uncorrelated states). They show this by numerical simulations of their QNN with a small number of neurons, which are taken to be qubit systems.

The results are scientifically sound, and the paper is very well presented. I believe that this is an important contribution for the QML and quantum algorithms community, specially since ML algorithms capable of dealing with quantum data are not abundant. Another potentially strong aspect of these results is that, at least in the experiments that the authors carried out, there seems to be no 'barren plateau' (a phenomenon where the training algorithm is incapable of driving the network to its optimal state for successful learning, if the network is initialized at random). If this would be a generic feature of the proposed architecture, it would significantly make it stand out among other proposed QNNs based on variational quantum circuits. Because of these reasons, I would recommend the publication of the manuscript in Nature Communications, provided the authors address satisfactorily the following comments.

1) The manuscript is missing some references that I deem relevant to set the work in context. In relation to the problem of learning an unknown quantum unitary transformation, this was first studied in PRA 81, 032324 (2010). The setting is seemingly different, but it actually contains the problem the authors address here: the unitary is given as a black box that can be used a finite number of times N . If the black box would be applied over a set of states $|\psi_{in,x}\rangle$, output states $V|\psi_{in,x}\rangle$ would be obtained, reproducing the training set that the authors of the manuscript consider. In this work, emphasis is put in the limitation of resources. That is, what is the most general form of (entangled) probes and circuit to test the black box for a given N , such that we learn as best as possible how to mimic the action of V on a new state. This is analogous to the

QNN reproducing V on the test states. This resource-centred perspective on learning unitaries has been recently generalized in PRL 122, 170502 (2019) to include probabilistic protocols.

2) To my knowledge, the CQ, QC, QQ terminology was first discussed in Aïmeur, Brassard, Gambs, "Machine learning in a quantum world", Advances in Artificial Intelligence, volume 4013 of Lecture Notes in Computer Science, 431-442 (2006). It is not original from the wikipedia article.

3) In relation to references on the QQ category. Quantum generalizations of ML-type problems that use "quantum data" as training information, i.e., quantum systems in specific states, had been already explored in several works prior to 2017. A selection of relevant ones could be:

Template matching: PRA 66, 022303 (2002)

Supervised learning: Sci. Rep. 2, 708 (2012), PRL 118, 190503 (2017), arXiv:0809.0444

Unsupervised classification: arXiv:1903.01391

Reinforcement learning: PRL 117, 130501 (2016)

A good review on the topic can be found in Rep. Prog. Phys. 81, 7 (2018).

4) In most of the above works, an emphasis is given to what can be done with a limited amount of quantum data. This makes sense, since the crucial difference between quantum and classical data is that the former cannot be cloned perfectly, or processed an arbitrary number of times without degrading it. Thus, ML algorithms are bound to incorporate this restriction in scenarios where resources are limited, and their design becomes highly nontrivial. The resulting algorithms have, in many cases, a radically different structure than their classical counterparts. In this manuscript, the authors simply say that enough copies of the training states are needed to "overcome projection noise" and properly train the QNN. This number is then not regarded as an important parameter of the problem. In particular, the report on numerical simulations provides the size of the training set, the number of rounds of the backpropagation algorithm, the learning rate, and the step size in the updates of the perceptron unitaries, but not the number of copies of the training set used in each round of training. This is a crucial parameter for any prospective experimental implementation of the QNN and should be reported.

My fear is that this number might be extremely high. According to appendix G, the derivative of the cost function is computed for each tunable parameter (and there are $\sum_l 4^{(m_{l-1}+1)} \times (\#perc \text{ in } l)$). Each such evaluation will need a number of copies of each state in the training set, and that is for a single training round. Is that correct? If so, the experimental implementation of the proposed QNN might just be infeasible for practical purposes for networks of more than a few neurons and layers. I want to clarify, though, that in any case this does not hinder the theoretical value of the work.

5) Building on the above point. In the appendices, the optimization of the cost function at each step is solved implicitly and the solution for updating the perceptron unitaries, the matrices K_j^l , is expressed in terms of the commutator M_j^l . This commutator is further simplified in a form that only depends on the states ρ_x^{l-1} (propagated from the input) and σ_x^l (back-propagated from the true output). I wonder if there is a way to directly evaluate this commutator and hence avoid computing a derivative for each parameter of the perceptron unitary U_j^l . For instance, imagine that for a given x we do state tomography in layer $l-1$ to obtain the density matrix ρ_x^{l-1} , and in layer l to obtain σ_x^l , and then compute classically M_j^l with the results. Could it be possible that such protocol is more efficient in terms of the number of copies of the pair $(|\phi_x\rangle, V|\phi_x\rangle)$? What would be the overall effect of finite statistics in the learning process?

6) A nuance. In appendix B it is mentioned that classical training data consists in instances of an unknown probability distribution, and that the natural quantization of probability distributions are density matrices. I agree with this reasoning. However, in their notion of quantum data, the

authors replace the classical instances by quantum states, not the underlying probability distribution itself. The example of learning a unitary transformation is not a good one to illustrate this point, as it is genuinely quantum. The distinction is clearer in a classification setting. Let two sets of N classical data points, labelled by 1 and 2, come from two different underlying probability distributions p_1 and p_2 . The quantization of this training set could be either two sets of N different quantum states that have been sampled from two different probability distributions over the Hilbert space, or N copies of a quantum state $|p_1\rangle$ and N copies of $|p_2\rangle$ (this is the approach followed in some of the references of point 3, see above). The latter quantization would be in line with the reasoning laid out appendix B, that is, quantizing the underlying probability distribution. I think it would be beneficial if these distinct notions of “quantum data” would be discussed in the manuscript.

7) The last sentence of appendix C mentions that it would be interesting to see which channels can be simulated by the QNN with perceptrons acting on $m+1$ qubits. Do you have an idea about how restrictive is such QNN with respect to an arbitrary quantum circuit of the same depth, at least intuitively? I think this is a very interesting point which would be nice to elaborate, if possible.

8) How would this QNN architecture work in a classification scenario? In the simplest case of two classes the output layer would be a single qubit, and the network should try to tailor its state towards either $|0\rangle$ or $|1\rangle$, depending on the class of the input. I’m aware that studying the behavior of the QNN for a classification problem probably implies quite some work, so I want to make clear that this is not a requisite for recommending publication. However, if it is easy or the authors have an idea of whether the same results (learning, generalization, and robustness) would hold also in a classification problem, it would be a nice addition to the paper.

9) On the above point. It could be that the good generalization performance that the QNN exhibits is a particularity of the problem of learning unitaries. The intuition is that a few random states are enough to pin down the action of the unknown unitary, leaving not much room for overfitting the training data. This could be very different in e.g. a classification problem, where the possible classes of states can be defined in an arbitrary or even pathological way, thus a particularly bad training set might be far away from representing faithfully the classes. Would you agree with this?

10) This is my personal opinion, but I think the title “Efficient learning of deep quantum neural networks” might be somewhat misleading. The efficiency, as I understand that the authors mean, comes from the necessary number of coherent qubits to control at each step of the algorithm. However, efficiency is also measured in terms of the number of training rounds, and (in the case of quantum data) in terms of the number of copies required, I would say. From these perspectives, the proposed algorithm is not so efficient.

11) By the end of the main text, the absence of a “barren plateau” is mentioned. This observation stems from the numerical simulations carried out by the authors. However, one could argue that the simulations performed are over rather small networks, where a barren plateau phenomenon might not yet manifest. On the other hand, if this absence would be a general feature of the proposed architecture, I would put a lot more emphasis in this point. Could you elaborate in this direction?

12) In a number of places a “sequel” is mentioned. I don’t know what this means. You do refer to the present manuscript and not to an upcoming second paper, right?

13) Appendix A.3, there is a mislabeled reference.

14) Appendix D.2, point II.2 and equations below: $m(l)$ should be m_l , for consistency.

15) Below equation (D.2). I think it would add clarity to define explicitly what “rest” means in the partial trace.

16) Above line 445, equation for $M_j^l(s)$. It is strange that the state in the first term of the commutator, ignoring the unitaries, is defined only over layers "in" and 1, but the state in the second term includes all layers with the identity $I_{in,hidden}$. It caused me some confusion for a bit. Maybe there is a better way of writing this.

17) When M_j^l is simplified (above line 448). "The formula for $M_j^l(s)$ in the training algorithm simplifies to...". It would be good to refer here to said formula with a reference.

18) Below line 457. "With probability 1..." This is not strictly true. It is with probability approaching or tending to 1.

19) At the bottom of page 16. "[...] we exploit the identity" That identity is derived (at least) from Schur lemma. It would be nice to the reader to mention it. Also, a reminder that the average is taken uniformly with respect to the Haar measure does not harm.

20) Appendix F.1. What is the dimension of the Hilbert space used for these numerics? It is not mentioned, should be.

21) Appendix F.3. Intuitively, adding layers increases the expressivity of the QNN. Can you see this in some way in your simulations? Is it actually wise to add intermediate layers for the problem of learning a unitary?

22) Appendix G.1. Point c. CSWAP. Some kets should be bras.

23) Appendix G.4. The notation x^α is extremely confusing. Before, x was the label for the states in the training. Also, α becomes a superindex of a Pauli operator in the same line. Furthermore, the x becomes X below G1 without warning. $\#perc$ should be properly defined too. Please change all this.

24) Appendix G.4. Could you clarify the argument why the cost function is always larger? In other words, I don't quite follow the step from 1st line to 2nd line in the equation in page 24.

Reviewer #2:

Remarks to the Author:

The authors propose a very natural definition of a quantum perceptron and derive a quantum neural network that can coherently learn unknown dynamics from labeled training data. The authors prove several results about their QNN framework, including an explicit derivation of the gradient for training purposes, and bolster their claims with numerical simulations that demonstrate some amount of robustness to corruption of the training labels. This is a very nice and simple result, and I'm a little surprised that someone hasn't suggested this definition sooner! I think it will certainly appeal to the broad readership of Nature Communications. The results are all correct as far as I have checked them.

I have a couple of optional comments for the authors to consider.

The cost function that the authors use is essentially one minus the average fidelity of the learned channel to the true channel. However, this measure can vary quite considerably from other metrics such as the diamond distance. In fact, it is possible to have a cost of ϵ and a diamond distance of order the square root of ϵ . Do the authors have any insights about 1) using a 'stronger' cost function such as one based on the diamond norm, or 2) would the results differ significantly using such a cost function? It might be worth a few sentences in the appendix somewhere to touch on this point since for quantum channels the diamond distance is very often

used as the canonical measure of distance between channels.

The notion of noise that the authors consider is natural if one has a functioning quantum computer with perfect logical qubits, but imperfect training data. However, the authors cite NISQ devices as some of the motivation for their work. In the context of NISQ devices, it is really the noise in the samples from the device that will decrease the contrast of the signal that will hurt the performance of the proposed QNN.

Consider the following example. The authors consider a model where instead of samples from labeled pairs (in, out), we sometimes get the pair (in, fake) instead. They show robustness in this case. Now instead we get samples from (in, out), but the random variable at the measurement is corrupted by noise with some probability p . Now the samples are accurate, but the signal is convolved through the noisy measurement channel. I don't think that the QNN will be robust to this type of noise because there is no way to distinguish noise in the unknown channel from the noisy measurements. In fact, this is a limitation of all of the proposed schemes for QML. I find it particularly annoying when QML and NISQ are uttered in the same breath and yet no one seems to care about noisy measurements leading to systematic bias in the results. I would greatly appreciate if the authors could find something intelligent to say about this in their article, even if it is only to acknowledge that this is presently a failure mode for their scheme, just to get some people aware of this issue.

typo: "To evaluate the benchmark the performance"

REPLY TO REVIEWERS

Authors: We thank each of the reviewers for carefully reading our manuscript and are especially grateful that they have committed so much of their time to reviewing this work. We greatly appreciate that they have made the effort to give comprehensive and constructive suggestions and are extremely grateful for their insightful comments and useful feedback. This has given us the opportunity to significantly improve the quality and the clarity of the paper. We address each of the reviewers' points below and describe the changes made to the manuscript based on their recommendations.

(The quoted text below from the reviewers is identical to their reports, except that mathematics has been put into latex form.)

Reviewer 1

1) The manuscript is missing some references that I deem relevant to set the work in context. In relation to the problem of learning an unknown quantum unitary transformation, this was first studied in PRA 81, 032324 (2010). The setting is seemingly different, but it actually contains the problem the authors address here: the unitary is given as a black box that can be used a finite number of times N . If the black box would be applied over a set of states $|\psi_{in,x}\rangle$, output states $V|\psi_{in,x}\rangle$ would be obtained, reproducing the training set that the authors of the manuscript consider. In this work, emphasis is put in the limitation of resources. That is, what is the most general form of (entangled) probes and circuit to test the black box for a given N , such that we learn as best as possible how to mimic the action of V on a new state. This is analogous to the QNN reproducing V on the test states. This resource-centred perspective on learning unitaries has been recently generalized in PRL 122, 170502 (2019) to include probabilistic protocols.

Authors: We are very grateful to the referee for bringing these references to our attention. We have added these references to the paper and explained their relevance. Although the problem the authors address in these papers is the same, there are some interesting differences to our work. In these papers, the goal is to learn an unknown quantum channel by accessing it a finite number of times and storing it in a state of a quantum memory such that it can be retrieved when needed. A crucial contrast to our setting is that we do not need a quantum memory to store the learned unitary, rather the parameters that characterize the unitary are stored in a classical register. Once these parameters have been learned, the unitary can be prepared and applied an arbitrary number of times.

2) To my knowledge, the CQ, QC, QQ terminology was first discussed in Aïmeur, Brassard, Gambs, "Machine learning in a quantum world", Advances in Artificial Intelligence, volume 4013 of Lecture Notes in Computer Science, 431-442 (2006). It is not original from the wikipedia article.

Authors: We thank the referee for pointing this out and have included the corresponding reference.

3) In relation to references on the QQ category. Quantum generalizations of ML-type problems that use “quantum data” as training information, i.e., quantum systems in specific states, had been already explored in several works prior to 2017. A selection of relevant ones could be:

Template matching: PRA 66, 022303 (2002)

Supervised learning: Sci. Rep. 2, 708 (2012), PRL 118, 190503 (2017), arXiv:0809.0444

Unsupervised classification: arXiv:1903.01391

Reinforcement learning: PRL 117, 130501 (2016)

A good review on the topic can be found in Rep. Prog. Phys. 81, 7 (2018).

Authors: Many thanks for providing these important contributions to the literature: we have included these references.

4) In most of the above works, an emphasis is given to what can be done with a limited amount of quantum data. This makes sense, since the crucial difference between quantum and classical data is that the former cannot be cloned perfectly, or processed an arbitrary number of times without degrading it. Thus, ML algorithms are bound to incorporate this restriction in scenarios where resources are limited, and their design becomes highly nontrivial. The resulting algorithms have, in many cases, a radically different structure than their classical counterparts. In this manuscript, the authors simply say that enough copies of the training states are needed to “overcome projection noise” and properly train the QNN. This number is then not regarded as an important parameter of the problem. In particular, the report on numerical simulations provides the size of the training set, the number of rounds of the backpropagation algorithm, the learning rate, and the step size in the updates of the perceptron unitaries, but not the number of copies of the training set used in each round of training. This is a crucial parameter for any prospective experimental implementation of the QNN and should be reported.

Authors: First of all, we thank the referee for highlighting this very important point. Regarding the numerical simulations in our work, it has to be clarified that they are classical simulations of the training of the QNN, they are not performed on a quantum computer. Hence, we can easily access many copies of the training set and do not have to worry about quantum projection noise. However, we agree with the referee that, for the quantum implementation, this is an important issue; we deal with this point below.

My fear is that this number might be extremely high. According to appendix G, the derivative of the cost function is computed for each tunable parameter (and there are $\sum_l 4^{(m_{l-1}+1)} \times (\#\text{perc in } l)$). Each such evaluation will need a number of copies of each state in the training set, and that is for a single training round. Is that correct? If so, the experimental implementation of the proposed QNN might just be infeasible for practical purposes for networks of more than a few neurons and layers. I want to clarify, though, that in any case

this does not hinder the theoretical value of the work.

Authors: The reviewer is indeed correct that the number of copies N_{copies} of each pair in the training set used in each round is large (we quantify this below). We obtain numbers on the order of 800 million copies. Admittedly this number is rather large. However, firstly, this is an extremely pessimistic overestimate, and, secondly, it is worth comparing this figure with the number of quantum circuit repetitions employed in a recently reported experiment involving the ‘‘Sycamore’’ quantum computer (see F. Arute et al., Nature **574**, 505–510 (2019)). Here it was reported that the experiment was repeated one million times taking 200 s. This is only two orders of magnitude away from the requirements of our method. The number of copies N_{copies} of each pair in the training set used in each round can be obtained from the following formula:

$$N_{\text{copies}} = n_{\text{proj}} \times n_{\text{params}}. \quad (1)$$

Here n_{proj} is the factor coming from repetition of measurements to reduce projection noise (i.e., estimate expectation values via measurement), and n_{params} is the total number of parameters in the network given by

$$\begin{aligned} n_{\text{params}} &= \sum_{l=1}^{L+1} \sum_{j=1}^{m_l} \# \text{ parameters}(U_j^l) \\ &= \sum_{l=1}^{L+1} \sum_{j=1}^{m_l} (4^{(m_{l-1}+1)} - 1) \\ &= \sum_{l=1}^{L+1} m_l \times (4^{(m_{l-1}+1)} - 1). \end{aligned} \quad (2)$$

U_j^l are the perceptron unitaries, and the index $L + 1$ refers to the outgoing layer (i.e., U_j^{out}). To get the second line, we used that the number of parameters in the perceptrons in this work is given by $4^{(m_{l-1}+1)} - 1$. Recall that m_l is the number of perceptron unitaries acting on perceptrons in layer $l - 1$ and layer l , and the -1 term occurs because the overall phase of the unitaries is unimportant. (This is the same as the formula given by the referee, except for the -1 .)

As an example, let us ask how many copies of each training pair we would need to perform the quantum training of the network in figure 2(b) in the paper. In that case, $n_{\text{params}} = 699$, and there were 300 rounds of training with 100 pairs. (Note that we actually need much fewer training pairs to train the network. The point in that part of the discussion was to test how robust the training is when some pairs are noisy. For our purposes 8 will suffice.) Then we have

that the total number of copies needed for training is

$$\begin{aligned}
N_{\text{total copies}} &= n_{\text{proj}} \times n_{\text{params}} \times n_{\text{pairs}} \times n_{\text{rounds}} \\
&= 500 \times 699 \times 8 \times 300 \\
&= 838,800,000.
\end{aligned}
\tag{3}$$

Here we chose $n_{\text{proj}} = 500$ to reduce the projection noise below ~ 0.03 . This used the bound on fluctuations discussed in appendix G1: the measurement used to estimate the fidelity estimates probabilities p with fluctuations given by $\sqrt{p(1-p)}/n_{\text{proj}} \leq 1/\sqrt{2n_{\text{proj}}}$. This estimate of $N_{\text{total copies}}$ is overkill to a large extent. If we wanted to actually trial such a process on a quantum computer, we wouldn't train for 300 rounds, and, especially for noisy systems without error correction, there would be less need to choose n_{proj} so large. This is because the system would be noisy anyway. The interesting point in that case would be to test how much the circuit can learn on a noisy system more as a proof of principle.

Another important point to bear in mind is that, in networks where we make an ansatz allowing reduced connectivity of the neural network, n_{params} can be reduced significantly.

Nevertheless, the reviewer is absolutely correct that the number of copies of the training data needed to train the network may be large, something which should be highlighted, so we have included this in the text (around line 176):

Furthermore, the number of copies per training round needed grows quickly with the number of neurons (linearly with the number of network parameters), i.e., $n_{\text{proj}} \times n_{\text{params}}$, where n_{proj} is the factor coming from repetition of measurements to reduce projection noise, and n_{params} is the total number of parameters in the network given by $\sum_{l=1}^{L+1} (4^{(m_{l-1}+1)} - 1) \times m_l$, where m_l is the number of perceptrons acting on layers $l-1$ and layer l , and the -1 term appears because the overall phase of the unitaries is unimportant. This means that in the near term, for large networks, only sparsely connected networks may be practical for experimental purposes. An exception would be if the problem being considered is such that the training data is easy to produce, e.g., if the output states are produced by allowing input states to thermalize by simply interacting with environment, thus producing the output states. Furthermore we add a detailed discussion in the new Appendix H.

5) Building on the above point. In the appendices, the optimization of the cost function at each step is solved implicitly and the solution for updating the perceptron unitaries, the matrices K_j^l , is expressed in terms of the commutator M_j^l . This commutator is further simplified in a form that only depends on the states ρ_x^{l-1} (propagated from the input) and σ_x^l (back-propagated from the true output). I wonder if there is a way to directly evaluate this commutator and hence avoid computing a derivative for each parameter of the perceptron unitary U_j^l . For instance, imagine that for a given x we do state tomography in layer $l-1$ to obtain the density matrix ρ_x^{l-1} , and in layer l to obtain σ_x^l , and then compute classically M_j^l with the results. Could it be possible that such protocol

is more efficient in terms of the number of copies of the pair $(|\phi_x\rangle, V|\phi_x\rangle)$?
 What would be the overall effect of finite statistics in the learning process?

Authors: This is an interesting idea, and there could be multiple ways to implement it. In the following, we discuss a few ideas. There could of course be better strategies that are not obvious to us. (Here, it's useful to look at point 3a in Figure 2 in the manuscript.)

(i) One option along these lines, is to use state tomography to find σ_x^l and ρ_x^{l-1} for a given l . From that we can find M_j^l , which will allow us to do the update. To do that we would need to simulate the evolution of $\rho_x^{l-1} \otimes |00\dots 0\rangle_l \langle 00\dots 0|$ and $I_{l-1} \otimes \sigma_x^l$ under the unitaries for those layers. However, the neural network should hopefully be useful in a regime where it cannot be easily classically simulated, so this would probably not be a feasible strategy.

(ii) We can consider an alternative use of the state tomography idea: to do the unitary updates in the second equation in point 3a and *then* do tomography to find the states, e.g., $\prod_{\alpha=j}^1 U_\alpha^l(\rho_x^{l-1} \otimes |00\dots 0\rangle_l \langle 00\dots 0|) \prod_{\alpha=1}^j U_\alpha^l$. But then we would still have to classically calculate the commutator, which would be a difficult task for such large matrices.

Still, we can count how many copies $N_{\text{copies}}^{\text{tom}}$ of each pair in the training set (per round of training) we would need in this case.

$$N_{\text{copies}}^{\text{tom}} = n_{\text{proj}} \times \sum_{l=1}^{L+1} m_l \times 2[(d_l \times d_{l-1})^2 - 1]. \quad (4)$$

Here n_{proj} is again the factor coming from repetition of measurements to reduce projection noise (i.e., estimate expectation values via measurement), which we also need when doing measurements for state tomography. Then we sum over the layers with the summand $m_l \times 2[(d_l \times d_{l-1})^2 - 1]$. Here $d_l \times d_{l-1}$ arises because the states we look at live in $d_l \times d_{l-1}$ -dimensional Hilbert spaces. The power of two follows from the usual reasoning that state tomography requires $O(d^2 - 1)$ measurements to characterize a state on a d -dimensional Hilbert space. The factor of 2 comes from the fact that we are doing this for two states, and finally the factor of m_l comes from the fact that there are m_l matrices M_j^l that we need to calculate for a given layer. Note that, if there were some additional information, we might be able to reduce the cost of the state tomography via, e.g., compressed sensing.

So the question now is how this value ($N_{\text{copies}}^{\text{tom}}$) compares to our value for N_{copies} from the previous answer. To make things simple, let's assume a network of qubits with constant width $m_l = m$ for all l . To compare $N_{\text{copies}}^{\text{tom}}$ and N_{copies} we can ignore factors of n_{proj} and just look at the summands (as the sums are over the same ranges: all layers). Then, because

$$m_l \times 2[(d_l \times d_{l-1})^2 - 1] = 2m(4^m - 1)^2, \quad (5)$$

is bigger than

$$(4^{m+1} - 1) \times m \quad (6)$$

for $m \geq 2$, we see that $N_{\text{copies}}^{\text{tom}} \geq N_{\text{copies}}$, so the state tomography trick doesn't help us in this case. A caveat is that with some useful ansatz about the states, some more specialized forms of state tomography may give us an advantage and $N_{\text{copies}}^{\text{tom}} \leq N_{\text{copies}}$, which could aid with training specialized networks. This is highly interesting and would be a good point for future work.

(iii) A final possibility is to use a modified version of a trick from S. Lloyd, M. Mohseni and P. Rebentrost, *Nat. Phys.* **10**, 631 (2014). Here, the idea is to encode the commutator of two states into the state of the system. Let's see how this works. Given two states ρ_A and σ_B of the same dimension, we evolve the state of A and B , i.e., $\rho_A \otimes \sigma_B$, via the unitary $\exp(-iS\pi/4)$, where S is the swap operator. Then we trace out system A :

$$\text{tr}_A[e^{-iS\pi/4} \rho_A \otimes \sigma_B e^{iS\pi/4}] = \frac{\rho_B + \sigma_B}{2} - \frac{i}{2}[\rho_B, \sigma_B], \quad (7)$$

where we used that $e^{-iSt} = \cos(t) - i \sin(t)S$, which follows because $S^2 = I$.

So we see that the commutator is encoded into the state of the B system. The update matrices K_j^l may be obtained from this expression by taking the partial trace, i.e., by measuring some local observables. To eliminate $(\rho_B + \sigma_B)/2$ we need to compute ρ_B and σ_B though. It might well be possible to make this approach coherent and eliminate completely the requirement that we have access to many copies of the training data; we will investigate this in a future paper.

We have added all of this discussion to the new appendix in the paper.

6) A nuance. In appendix B it is mentioned that classical training data consists in instances of an unknown probability distribution, and that the natural quantization of probability distributions are density matrices. I agree with this reasoning. However, in their notion of quantum data, the authors replace the classical instances by quantum states, not the underlying probability distribution itself. The example of learning a unitary transformation is not a good one to illustrate this point, as it is genuinely quantum. The distinction is clearer in a classification setting. Let two sets of N classical data points, labelled by 1 and 2, come from two different underlying probability distributions p_1 and p_2 . The quantization of this training set could be either two sets of N different quantum states that have been sampled from two different probability distributions over the Hilbert space, or N copies of a quantum state $|p_1\rangle$ and N copies of $|p_2\rangle$ (this is the approach followed in some of the references of point 3, see above). The latter quantization would be in line with the reasoning laid out appendix B, that is, quantizing the underlying probability distribution. I think it would be beneficial if these distinct notions of "quantum data" would be discussed in the manuscript.

Authors: We add around line 165:

There are two notions of "quantum data": Two sets of N classical data points come from two different underlying probability distributions p_1 and p_2 . The quantisation of this training set could be either two sets of N different quantum states that have been sampled from two different probability distributions over the Hilbert space, or N copies of a quantum state. We assume the second version.

7) The last sentence of appendix C mentions that it would be interesting to see which channels can be simulated by the QNN with perceptrons acting on $m+1$ qubits. Do you have an idea about how restrictive is such QNN with respect to an arbitrary quantum circuit of the same depth, at least intuitively? I think this is a very interesting point which would be nice to elaborate, if possible.

Authors: With regard to qubits vs. qudits, we recently discovered that qubit perceptrons are also universal for quantum computation. This argument has been included in Appendix C.

In recent investigations of the quantum neural network we have developed a training algorithm which is able to train the QNN with respect training data with non-pure output states. This algorithm is very similar to the one presented in the paper and is based the cost function being an average of Hilbert-Schmidt norms, i.e. for $(\rho_x^{\text{in}}, \rho_x^{\text{out}})_{x=1, \dots, N}$ a set of training pairs and \mathcal{E}_{QNN} being the channel corresponding to the QNN this cost function is given by

$$C = \sum_{x=1}^N \|\rho_x^{\text{out}} - \mathcal{E}_{\text{QNN}}(\rho_x^{\text{in}})\|_2^2.$$

Using this new training algorithm we were able train the network with respect training sets $(\rho_x^{\text{in}}, T(\rho_x^{\text{in}}))_{x=1, \dots, N}$, with T being a randomly generated channel. From the numerics we have done so far the training seems to be successful as the network was able to perfectly learn with respect to such training sets.

8) How would this QNN architecture work in a classification scenario? In the simplest case of two classes the output layer would be a single qubit, and the network should try to tailor its state towards either $|0\rangle$ or $|1\rangle$, depending on the class of the input. I'm aware that studying the behavior of the QNN for a classification problem probably implies quite some work, so I want to make clear that this is not a requisite for recommending publication. However, if it is easy or the authors have an idea of whether the same results (learning, generalization, and robustness) would hold also in a classification problem, it would be a nice addition to the paper.

Authors: The referee is proposing an interesting application of our quantum network architecture, which is one that we are actually currently studying. In our investigations, we have found that for the simple task that the referee is proposing, namely having two classes of states which we want to distinguish, the network is indeed able to train to the optimal value of the cost function. We have

added a more detailed explanation and a plot of the training behavior of the network to Appendix F.4. A much deeper investigation of how the network performs in classification tasks is part of a future publication.

9) On the above point. It could be that the good generalization performance that the QNN exhibits is a particularity of the problem of learning unitaries. The intuition is that a few random states are enough to pin down the action of the unknown unitary, leaving not much room for overfitting the training data. This could be very different in e.g. a classification problem, where the possible classes of states can be defined in an arbitrary or even pathological way, thus a particularly bad training set might be far away from representing faithfully the classes. Would you agree with this?

Authors: We agree with the intuition proposed by the referee that a few random states are enough to pin down the action of the unknown unitary (this will be the subject of an upcoming paper on a quantum generalisation of the no free lunch theorem). About general classification problems: we have investigated the optimality of such schemes both via SDPs and QNNs. We have found (in preparation), somewhat surprisingly, that in practice QNNs give competitive results. We added a section in Appendix F.4.

10) This is my personal opinion, but I think the title “Efficient learning of deep quantum neural networks” might be somewhat misleading. The efficiency, as I understand that the authors mean, comes from the necessary number of coherent qubits to control at each step of the algorithm. However, efficiency is also measured in terms of the number of training rounds, and (in the case of quantum data) in terms of the number of copies required, I would say. From these perspectives, the proposed algorithm is not so efficient.

Authors: We thank the reviewer for pointing out the misleading title and changed it to ‘Training deep quantum neural networks’.

11) By the end of the main text, the absence of a “barren plateau” is mentioned. This observation stems from the numerical simulations carried out by the authors. However, one could argue that the simulations performed are over rather small networks, where a barren plateau phenomenon might not yet manifest. On the other hand, if this absence would be a general feature of the proposed architecture, I would put a lot more emphasis in this point. Could you elaborate in this direction?

Authors: We are grateful to the referee for emphasising this point. There are two key reasons we believe that our QNNs will not exhibit a barren plateau. According to J. R. McClean, S. Boixo, V. N. Smelyanskiy, R. Babbush, and H. Neven, *Nat. Commun.* **9**, 4812 (2018), “The gradient in a classical deep neural network can vanish exponentially in the number of layers [...], while in the a quantum circuit is exponentially small in the number of qubits,” This point does not apply to our QNNs because the gradient of a weight in the

QNN doesn't depend on all the qubits but rather only on the number of paths connecting that neuron to the output, just as it does classically. (This is best observed in the Heisenberg picture.) Thus, indeed, the gradient vanishes exponentially in the number of layers, but not in the number of qubits, just as it does classically. Secondly, our cost function differs from that of (J. R. McClean, S. Boixo, V. N. Smelyanskiy, R. Babbush, and H. Neven, Nat. Commun. 9, 4812 (2018).): they consider energy minimisation of a local hamiltonian, whereas we consider a quantum version of the risk function. Our quantity is not local, and this means that Levy's lemma-type argumentation doesn't directly apply.

We have added this discussion to the paper around line 274.

12) In a number of places a *sequel* is mentioned. I don't know what this means. You do refer to the present manuscript and not to an upcoming second paper, right?

Authors: We replaced the word with synonyms for better understanding.

13) Appendix A.3, there is a mislabeled reference.

Authors: Thanks for spotting this error, we have removed this.

14) Appendix D.2, point II.2 and equations below: $m(l)$ should be m_l , for consistency.

Authors: We changed $m(l)$ to m_l for consistency everywhere.

15) Below equation (D.2). I think it would add clarity to define explicitly what *rest* means in the partial trace.

Authors: Added the sentence: Note that *rest* in tr_{rest} refers to the complement of $\{\alpha_1, \dots, \beta\}$.

16) Above line 445, equation for $M_j^l(s)$. It is strange that the state in the first term of the commutator, ignoring the unitaries, is defined only over layers "in" and 1, but the state in the second term includes all layers with the identity $I_{\text{in,hidden}}$. It caused me some confusion for a bit. Maybe there is a better way of writing this.

Authors: This is indeed confusing and the way it is written in the equation is actually not quite correct. The state in the first term of the commutator should actually be a state of all the hidden layers and the output layers. We changed this accordingly.

17) When M_j^l is simplified (above line 448). *The formula for $M_j^l(s)$ in the training algorithm simplifies to....* It would be good to refer here to said formula with a reference.

Authors: We added the reference to the equation for $M_j^l(s)$.

18) Below line 457. *With probability 1...* This is not strictly true. It is with probability approaching or tending to 1.

Authors: We thank the reviewer for mentioning that this part is not written clearly enough and can lead to confusion. The intuition one needs for this part is that two randomly chosen vectors never point in the same direction. Therefore, we changed the part in the paper to the following:

With probability 0 any randomly chosen subset of D of the states $|\phi_x\rangle$ will be linearly dependent. Thus the first $n < D$ states $|\phi_x\rangle$ span, with probability 1, an n -dimensional subspace $\mathcal{K} \subset \mathcal{H} \cong \mathbb{C}^D$ which is unitarily mapped by V onto an n -dimensional subspace \mathcal{L} .

19) At the bottom of page 16.[...] *we exploit the identity* That identity is derived (at least) from Schur lemma. It would be nice to the reader to mention it. Also, a reminder that the average is taken uniformly with respect to the Haar measure does not harm.

Authors: Many thanks for pointing this out. Bases on the suggestion we added the following sentence:

Note that this is derived from Schur lemma and the average is taken uniformly with respect to the Haar measure.

20) Appendix F.1. What is the dimension of the Hilbert space used for these numerics? It is not mentioned, should be.

Authors: Thank you for pointing out that this information is missing. The dimension that is used for these numerics is 2^n , where n is the number of input (respectively, output) qubits. We have added this information to the figures.

21) Appendix F.3. Intuitively, adding layers increases the expressivity of the QNN. Can you see this in some way in your simulations? Is it actually wise to add intermediate layers for the problem of learning a unitary?

Authors: Many thanks for addressing this important point. For the problem of learning a unitary it was always sufficient to have just one input and one output layer and no hidden layer to perfectly learn (i.e. cost function equal to one) with respect to a given training set. However, although using additional hidden layers was not necessary for this particular task, bigger networks also did not cause any problems for perfectly learning a unitary.

From recent investigations of the learning capability of the presented QNN we however know that there are tasks where deeper networks perform better than smaller ones. This often occurs when the dimension of the output space is bigger than the dimension of the input space, e.g. if you want to learn an isometry. Here one will often only get perfect training if one chooses a right network architecture with certain hidden layers. Similar effects can be observed when one uses the network to learn random channels as mentioned above (point 7).

We added additional thoughts around line 427.
Investigating these observations is work in progress and will be presented in future publication.

22) Appendix G.1. Point c. CSWAP. Some kets should be bras.

Authors: Thanks for noticing this, we have corrected the equation.

23) Appendix G.4. The notation x^α is extremely confusing. Before, x was the label for the states in the training. Also, alpha becomes a superindex of a Pauli operator in the same line. Furthermore, the x becomes X below G1 without warning. $\#perc$ should be properly defined too. Please change all this.

Authors: We are grateful to the referee for pointing out this possible point of confusion. We have revised the notation to improve clarity and changed x^α and X^α to y^μ and also explained $\#perc$ in an additional sentence.

24) Appendix G.4. Could you clarify the argument why the cost function is always larger? In other words, I don't quite follow the step from 1st line to 2nd line in the equation in page 24.

Authors: We changed the formulas in that argument to make it more explicit that the cost function always gets larger because the dominating term in the Taylor expansion is non-negative as it is quadratic.

Reviewer 2

The authors propose a very natural definition of a quantum perceptron and derive a quantum neural network that can coherently learn unknown dynamics from labeled training data. The authors prove several results about their QNN framework, including an explicit derivation of the gradient for training purposes, and bolster their claims with numerical simulations that demonstrate some amount of robustness to corruption of the training labels. This is a very nice and simple result, and I'm a little surprised that someone hasn't suggested this definition sooner! I think it will certainly appeal to the broad readership of Nature Communications. The results are all correct as far as I have checked them.

I have a couple of optional comments for the authors to consider.

The cost function that the authors use is essentially one minus the average fidelity of the learned channel to the true channel. However, this measure can vary quite considerably from other metrics such as the diamond distance. In fact, it is possible to have a cost of epsilon and a diamond distance of order the square root of epsilon. Do the authors have any insights about 1) using a "stronger" cost function such as one based on the diamond norm, or 2) would the results differ significantly using such a cost function? It might be worth a few sentences in the appendix somewhere to touch on this point since for quantum channels the diamond distance is very often used as the canonical measure of

distance between channels.

Authors: We thank the referee for emphasising this important point. We do agree that the diamond norm is operationally better suited for many quantum information processing tasks. Unfortunately we do not know of an efficient quantum approach to simulating this norm, and were unable to simulate learning according to this norm. We could simulate learning via the diamond norm classically using convex programming techniques (Ben-Aroya, Avraham, and Amnon Ta-Shma. "On the complexity of approximating the diamond norm." arXiv preprint arXiv:0902.3397 (2009).), but this does not easily extend to the quantum setting. Our justification for the cost function we use is that it is a direct generalisation of the risk function considered in training classical deep networks and we can efficiently simulate it. We have updated the paper to clarify this.

The notion of noise that the authors consider is natural if one has a functioning quantum computer with perfect logical qubits, but imperfect training data. However, the authors cite NISQ devices as some of the motivation for their work. In the context of NISQ devices, it is really the noise in the samples from the device that will decrease the contrast of the signal that will hurt the performance of the proposed QNN.

Consider the following example. The authors consider a model where instead of samples from labeled pairs (in, out), we sometimes get the pair (in, fake) instead. They show robustness in this case. Now instead we get samples from (in, out), but the random variable at the measurement is corrupted by noise with some probability p . Now the samples are accurate, but the signal is convolved through the noisy measurement channel. I don't think that the QNN will be robust to this type of noise because there is no way to distinguish noise in the unknown channel from the noisy measurements. In fact, this is a limitation of all of the proposed schemes for QML. I find it particularly annoying when QML and NISQ are uttered in the same breath and yet no one seems to care about noisy measurements leading to systematic bias in the results. I would greatly appreciate if the authors could find something intelligent to say about this in their article, even if it is only to acknowledge that this is presently a failure mode for their scheme, just to get some people aware of this issue.

Authors: We agree with the referee that we have not addressed this problem adequately in our paper. We now shortly mention that one would have to consider this kind of noise within the network itself for NISQ devices in the introduction and discuss it further near the end of the paper (around line 319): We have also performed extensive additional simulations of learning under more realistic noise and found that the QNN still learns.

A crucial problem that has to be taken into account with regard to NISQ devices is the inevitable noise within the device itself. Interestingly, we have obtained numerical evidence that, for approximate depolarising noise, QNNs are robust (see inset of Fig. 3).

typo: "To evaluate the benchmark the performance"

Authors: We have corrected the typo.

Reviewers' Comments:

Reviewer #1:

Remarks to the Author:

All my comments have been addressed by the authors satisfactorily, hence I recommend publication. I would only suggest the authors to implement the following two changes:

- Ref. [24] has been published in the meantime. Please update to Phys. Rev. X 9, 041029 (2019).
- Regarding the different notions of quantum data. I'm not sure the added paragraph adds much clarity. In my report I used the example of classification with two probability distributions just as an illustration. In general, it need not be two distributions. The crucial point is which element of a classical scenario is identified with a quantum state: either each classical sample of an unknown underlying probability distribution is replaced by a different quantum state (hence in the quantum scenario the underlying distribution will be a distribution over quantum states), or the distribution itself is the quantum state (in this case, it is justified to say that N samples will correspond to N identical quantum states). I suggest the authors to improve the phrasing of this argument.

Reviewer #2:

Remarks to the Author:

The authors have addressed all of my comments and I think that this manuscript is now suitable for publication in Nature Comms.

REPLY TO REVIEWERS

Reviewer 1

1) Ref. [24] has been published in the meantime. Please update to Phys. Rev. X 9, 041029 (2019).

Authors: We thank the referee for pointing this out and have updated the corresponding reference.

2) Regarding the different notions of quantum data. I'm not sure the added paragraph adds much clarity. In my report I used the example of classification with two probability distributions just as an illustration. In general, it need not be two distributions. The crucial point is which element of a classical scenario is identified with a quantum state: either each classical sample of an unknown underlying probability distribution is replaced by a different quantum state (hence in the quantum scenario the underlying distribution will be a distribution over quantum states), or the distribution itself is the quantum state (in this case, it is justified to say that N samples will correspond to N identical quantum states). I suggest the authors to improve the phrasing of this argument.

Authors: We thank the referee for clarifying this and have revised the paragraph about quantum data, which now says:

Now that we have an architecture for our QNN we can specify the learning task. Here, it is important to be clear about what part of the classical scenario we quantize. One possibility is to replace each classical sample of an unknown underlying probability distribution by a different quantum state. Hence, in the quantum setting, the underlying probability distribution will then be a distribution over quantum states. The second possibility is to identify the distribution itself with a quantum state, which we assume in this work, in which case it is justified to say that N samples correspond to N identical quantum states.